# Hikikomori: A Scientometric Review of 20 Years of Research

**DOI:** 10.3390/ijerph20095657

**Published:** 2023-04-27

**Authors:** Michelle Jin Yee Neoh, Alessandro Carollo, Mengyu Lim, Gianluca Esposito

**Affiliations:** 1Psychology Program, School of Social Sciences, Nanyang Technological University, Singapore 639818, Singapore; 2Department of Psychology and Cognitive Science, University of Trento, 38068 Rovereto, Italy

**Keywords:** document co-citation analysis, country analysis, hikikomori, social withdrawal, scientometrics

## Abstract

The Japanese term *hikikomori* was first used to describe prolonged social withdrawal in the 1990s. Since then, research across the world have reported similar prolonged social withdrawal in many countries outside Japan. This study systematically analyses the evolution of literature on *hikikomori* in the past 20 years to gain a better understanding of the development of the knowledge base on *hikikomori* since it garnered attention in Japan. Findings from the scientometric review indicate many perspectives on the etiology of *hikikomori* including cultural, attachment, family systems and sociological approaches. However, similarities with modern type depression, a novel psychiatric syndrome, have been proposed and there are signs of a recent paradigm shift of *hikikomori* as a society-bound syndrome rather than a cultural-bound syndrome unique to Japan. As research into *hikikomori* continues to grow, results from the review also highlight the need for a more universally shared definition of *hikikomori* in order to better consolidate cross-cultural research for meaningful and valid cross-cultural comparisons which can help to promote evidence-based therapeutic interventions for *hikikomori*.

## 1. Introduction

The term *hikikomori* was first used to refer to an individual who has stopped going to school or work and remained at home for a duration greater than six months by Tamaki Saito in 1998 [1]. Although *hikikomori* has not been recognised as a formal psychiatric diagnosis, the Ministry of Health, Labour and Welfare in Japan released guidelines for the assessment and treatment of *hikikomori* in collaboration with a research group in 2010 [2]. Two typologies of *hikikomori* have also been proposed: (i) primary *hikikomori* where the individual did not have any comorbid psychiatric disorder causing the social withdrawal and (ii) secondary *hikikomori* where the social withdrawal can be attributed to a psychiatric disorder [3,4]. Based on an epidemiological survey conducted in Japan, the lifetime prevalence of *hikikomori* was reported to be 1.2%, and was more commonly reported in males and individuals in their 20s [5]. Similarly, another review reported the prevalence to be 0.9% to 3.8% in Japan based on the results of three population studies [6]. The *hikikomori* phenomenon has become of increasing concern in the face of what government officials have termed the 8050 crisis, referring to *hikikomori* who are turning 50 while their parents reach their 80s, creating an unsustainable situation where these elderly parents may begin to lose the ability to support themselves and their children [7]. For example, statistics released by the Japanese Cabinet Office in 2019 reported an estimated 610,000 *hikikomori* aged between 40 and 65 [8]. Hence, this situation highlights the dire need for research into better understanding *hikikomori* and formulating interventions to promote the reintegration of *hikikomori* into society, especially considering that these figures are likely to be a conservative estimate [2].

The causes and risk factors for *hikikomori* are not well understood although many studies have highlighted the male gender and insecure attachment [9]. The co-dependency between parent and child characteristic of Japanese parent-child relationships—termed *amae* in Japanese—has been hypothesised to enable the development of *hikikomori*. Furlong [10] also discusses a number of proposed precipitating sociocultural factors in Japan including the Japanese family context characterised by a tendency of overprotection and indulgence of children, the high pressure and stakes of the Japanese education system and the rapidly contracting economy and labour market in the 1990s. More recently, Kato et al. [11] has proposed a multidimensional model of *hikikomori* with a biopsychosocial perspective, which considers both psychiatric and non-psychiatric *hikikomori* conditions.

### Aims of the Study

In recent decades, *hikikomori* has garnered increasing attention not only in Japan but also globally, where countries cross Asia, Europe and North America such as South Korea, China, Italy, France, Spain and the United States of America have reported similar cases of prolonged social withdrawal [12,13,14,15,16,17]. Hence, this study aimed to identify key publications and trends in the research on *hikikomori*, the focus of these publications and gaps in the literature, which are able to contribute to greater insight into directions for future research. Research on *hikikomori* can also benefit from a consolidation of research conducted across the world to facilitate valid cross-cultural comparisons and developing a clearer and more comprehensive understanding of the etiology of *hikikomori*. In the current study, a scientometric approach to reviews will be adopted, which combines elements from scientific mapping and bibliometric analysis. [18]. Scientometrics has proved useful in reviewing the literature in fields such as neuroscience and clinical psychology (e.g., [19,20,21]). In the current work, an analysis of the references and relevance of publications in the existing literature was conducted using a document co-citation analysis (DCA) [20,22,23]. A country analysis was also conducted in order to identify leading countries contributing to the research and knowledge base on *hikikomori*.

## 2. Materials and Methods

As done in Neoh et al. [21,24], the following search string TITLE-ABS-KEY (“hikikomori”) was used in the download of publications from Scopus according to standard and established scientometric procedures outlined in Chen [22]. The Scopus database was selected due to its coverage of a greater number of indexed journals and recent documents [25]. A search conducted on 9 November 2022 revealed that there was a total of 302 documents published from 1 January 2002 onward.

### 2.1. Data Import on CiteSpace

CiteSpace software (Version 6.1.R2 and version 6.1.R6) [26] was used to conduct the scientometric analysis in this study. Results presented in the main manuscript were obtained using the newest version of the software, while results obtained with the verison 6.1.R2 are presented in the Appendix A. The same procedures as in previous studies from our team [24,27] were followed to download the articles. The downloaded articles from Scopus were imported into the software where there were 11,373 (97.36%) valid references out of a total of 11,681 references cited by the 302 articles (Figure 1).

### 2.2. Document Co-Citation Analysis (DCA), Country Analysis and Parameter Optimisation

By conducting a DCA which is based on the frequency of papers being cited together in source articles [28], main research domains in the literature can be identified, including the construction of a network consisting of documents that have a high frequency of being cited together along with documents citing them. When articles are frequently cited together, it can be assumed that they reflect shared research trends and intellectual domains [23,29].

A parameter optimisation was conducted for the purpose of obtaining a balanced network of documents. This was achieved through the computation and comparison of multiple DCAS, which differed in terms of the settings for one of the three node selection criteria; g-index, TOP *N*, TOP *N*%, as done in [19,29,30,31,32,33,34]. The node selection criteria are predetermined parameters that define the criterion for the selection of articles for inclusion into the network, ultimately determining the generation of the final network. The g-index is a measure of the citation scores of the top publications by an author [35,36]. Given an article list ranked according to the number of received citations in descending order, the g-index is the largest number where the total number of citations received by the top g articles equals to at least g^2^ [35]. TOP *N* and TOP *N*% are criteria used to select the *N* and *N*% most cited within a particular time slice, as network nodes respectively [22]. In this study, we have set the duration of the time slice to be 1 year, which means that the node selection criteria and scale factors were applied on a year-by-year basis to retrieve the maximum amount of information from the data sample.

To enable users to control the overall size of the final network, CiteSpace uses the scale factor to implement a modified version of the node selection criteria. Scale factor values refer to the selected numeric value which are employed as thresholds for the node selection criteria. For example, with *k* = 1, the standard g-index is used. Conversely, higher values of *k* correspond to a higher number of included documents. Therefore, the scale factor sets the threshold of the criteria. In order to generate the final optimal network, a number of DCAs with varying node selection criteria and scale factor values were computed [32]. A comparison of the following DCAs and their node selection criteria and scale factor values was carried out: g-index with scale factor *k* set at 25, 50 and 75, TOP *N* with scale factor *N* set at 50, 100 and TOP *N*% with scale factor *N* set at 10. The structural metrics, number of included nodes and identified clusters were compared for the determination of the node selection criteria and scale factor to be employed for the generation of the final article network. Ultimately, a DCA with g-index with the scale factor *k* set at 50 was used.

Alongside the DCA, the involvement of countries in work on *hikikomori* was investigated through a country analysis. For the country analysis, g-index with the scale factor *k* set at 25 was the optimal parameter for the network generation. The country analysis relies on country information retrieved from the authors’ affiliations in the citing documents.

### 2.3. Metrics

Structural and temporal metrics are used to describe the CiteSpace results. Structural metrics include the following: (i) *modularity-Q*, (ii) *silhouette scores* and (iii) *betweenness centrality*. The modularity-Q value is indicative of the degree of decomposition into single groups of nodes, otherwise known as as modules or clusters [37]. Modularity-Q values have a range from 0 to 1 where higher modularity-Q values are an indicator of a well-structured network [23]. Silhouette scores measures inner consistency of modules in terms of internal cohesion and separation from other clusters [38]. The values range from −1 to +1, with higher values being indicative of greater separation from other modules and internal consistency [39]. Betweenness centrality is a measure with values ranging from 0 to 1, which is representative of the degree to which a node serves as a connection between an arbitrary node pair in the network [22,40]. Scores closer to 1 are usually observed for high-impact works in the scientific literature [37].

Temporal metrics include (i) *citation burstness* and (ii) *sigma*. The Kleinberg’s algorithm [41] is used in the calculation of citation burstness and indicates an abrupt increase in the number of citations of an article within a particular time frame [42]. The equation (centrality + 1)^burstness^ was used for the calculation of the sigma value. Sigma allows the consideration of both structural and temporal properties of a node, for this reason, it indicates a document’s novelty and influence on the overall network [23]. Sigma was introduced by Chen et al. [43].

For the examination of the overall configuration of the network and identified clusters, modularity-Q and silhouette scores were used. For the examination of single node characteristics, betweenness centrality and the temporal metrics were used.

## 3. Results

### 3.1. Structural Metrics

The DCA resulted in the generation of an optimised network made up of 992 nodes with 3488 links, which means there was an average of 3.52 connections with other references for each node. The network had a modularity-Q index of 0.8854 and a mean silhouette score of 0.9435, indicating high divisibility of the network into homogeneous clusters. This network resulted to be the most balanced as compared to the ones generated with g-index with scale factor *k* set at 25 (nodes = 567; links = 2110; modularity-Q = 0.7665; mean silhouette score = 0.9296; major clusters = 12) and 75 (nodes = 1332; links = 4644; modularity-Q = 0.9246; mean silhouette score = 0.9522; major clusters = 17), TOP *N* with scale factor *N* set at 50 (nodes = 218; links = 949; modularity-Q = 0.5879; mean silhouette score = 0.8724; major clusters = 7), 100 (nodes = 218; links = 949; modularity-Q = 0.5879; mean silhouette score = 0.8724; major clusters = 7) and TOP *N*% with scale factor *N* set at 10 (nodes = 218; links = 949; modularity-Q = 0.5879; mean silhouette score = 0.8724; major clusters = 7).

### 3.2. Thematic Clusters

16 major clusters were identified in total (Figure 2, Table 1). Based on their metrics, these 16 major clusters were automatically selected by CiteSpace from the total sample of 135 clusters. CiteSpace also automatically generates cluster labels using the log-likelihood ratio method (LLR Label). After a qualitative inspection of these CiteSpace-generated labels, the LLR labels were amended manually where necessary to better reflect the theme of the cluster, where the manually generated labels can be found in the table (Suggested Label). Cluster #0 was the largest cluster, consisting of 102 nodes with a silhouette score of 0.92, where the mean year of publication of the constituent references was 2016. The cluster was manually labelled “Clinical features”. Cluster #1 was the next largest cluster, consisting of 82 nodes with a silhouette score of 0.88, where the mean year of publication of the constituent references was 2015. The cluster was manually labelled “Risk factors”. Cluster #3 was the third largest cluster, consisting of 49 nodes with a silhouette score of 0.964, where the mean year of publication of the constituent references was 2007. The cluster was manually labelled “Family factors”.

### 3.3. Citation Burstness

14 documents exhibited a citation burst in total (Table 2), after duplicates of the same documents were removed from the results of the citation burstness table. The strongest citation burst was observed in the article authored by Teo [44] with a score of 6.72, with the burst beginning in 2013 to 2018. There were 3 articles with the longest burst duration of 5 years: Teo [44], Teo and Gaw [45] from 2013 to 2018, and Saito and Angles [1] from 2017 to 2022. The highest sigma value of 1.36 was observed in the article authored by Teo [44]. The betweenness centrality values of these documents appear to be low, suggesting there is no document that is highly influential in the overall network of documents. Low betweenness centrality values suggest that nodes are all homogeneously connected to each other, with no document being a common “bridge” when moving from a node to another. The results of this metric suggest that no node, if removed, would change the overall configuration of the network.

### 3.4. Country Analysis

The country analysis generated a network with 47 nodes (i.e., countries) and 97 links. The same results were observed with *k* set at 50 and 75, while smaller networks were observed when using TOP *N* with *N* set at 50 and 100 (nodes = 16; links = 23) and TOP *N*% with *N* set at 10 (nodes = 4; links = 2). In the optimal network, a total of 5 countries showed a citation burst when γ = 0.60. The parameter γ modulates the sensitivity of the node’s burst detection [41]. The lower the parameter, the easier it is for a document to obtain a citation burst. Although the default value of gamma is 1, we lowered the threshold in order to obtain a good sample of documents with a burst.The five countries with a citation burst were France, the United States, Switzerland, Hong Kong and Singapore (Table 3).

### 3.5. Differences between Versions of Cite Space

In the current manuscript, we used two different versions of CiteSpace. The structural properties of the generated DCA network were similar between the two versions. However, the updated version allowed the identification of 16 thematic clusters in the literature as opposed to the 14 identified with the older version of the software. When comparing the clusters obtained with the two versions of the software, themes were largely stable, with some differences that emerged mainly in regard to smaller thematic clusters. The same applies to the burst and country analyses. Negligible differences in terms of clustering and individual metric values are due to continuous improvements to the software’s algorithm.

## 4. Discussion

In this section, we will discuss clusters chronologically from the oldest to the most recent mean year of publication. The citing articles and cited references will be discussed, where the main citing articles in each cluster will be reported along with and their coverage and Global Citing Score (GCS). A list of cited references in each cluster can be found in the Appendix A. Coverage refers to the number of articles in the cluster that were cited by the citing article and GCS refers to the total number of citations received by a paper as indexed on Scopus.

### 4.1. Cluster #15: The Role of Society

Cluster “The role of society” included documents with an average year of publication in 2001. The major citing documents in the cluster was authored by Sakamoto et al. [52] (coverage = 14; GCS = 72). Particularly, the document presents a case of *hikikomori* in Oman, suggesting that the social commonalities between Omani and Japanese societies could reinforce the typical *hikikomori* behaviours.

### 4.2. Cluster #3: Family Factors

The articles citing the most references in the cluster in Cluster “Family factors” were authored by Toivonen [53] covering 15 articles and GCS of 17, Teo and Gaw [45] covering 12 articles and GCS of 130 and Umeda et al. [54] covering 9 articles and GCS of 39 (Table 4). In the literature, the family environment (e.g., parenting style, socioeconomic status) has often been cited as a risk factor for the development of *hikikomori* as well as a target for intervention. For example, the prevalence of *hikikomori* appears to be higher in middle and upper class families. It has also been argued that *amae*, a doting, indulgent and protective parenting style characteristic of parenting in Japan, may foster dependency of children on parents, which continues to be acceptable even as the child transitions to adulthood [55], leading to the acceptance of parents that their children remain at home for extended periods of time [11,12]. However, there is little systematic and empirical evidence supporting this notion as noted by Umeda et al. [54]. The role of the Japanese family context in *hikikomori* is discussed in a number of the cited references [10,56]. Other cited references included case studies where aspects of the family environment are highlighted but not statistically analysed as a factor for *hikikomori* development. For example, Hattori [57] detailed case studies of 35 *hikikomori* patients, reporting only the frequencies of symptoms and the treatment plans for these case studies. Notably, Umeda et al. [54] reported a higher likelihood of the occurrence of *hikikomori* in families where parents had a higher level of education. It was proposed that higher parental education levels imply higher incomes, possibly indicating greater financial ability to sustain the or higher parental expectations placed on children. Hence, it is clear that although parenting and the family environment have been proposed to be potentially significant contributors in the development of *hikikomori* even in the early stages of research, more well designed and empirical findings are needed to clarify this relationship to identify those at risk and inform the design of evidence-based interventions.

### 4.3. Cluster #9: Japanese Youth Labels

The articles citing the most references in the cluster in cluster “Japanese youth labels” were authored by Toivonen and Imoto [59] covering 15 articles and GCS of 9 and Abel [60] covering 6 articles and GCS of 9. This cluster highlights the nature of *hikikomori* as one of many “youth problem” (*wakamono mondai*) labels in Japan, including *otaku*, *NEET* (Not in Education, Employment or Training) and *freeter* (individuals who are unemployed or do not have full-time employment), which are recognizable and widely used. As discussed in Toivonen and Imoto [59], the use of such labels may not only pose a hindrance to research design due to the associated assumptions but often contributes to the phenomenon of “moral panic” in society, which can divert from targeted and effective interventions for *hikikomori*. Moreover, both Toivonen and Imoto [59] and Abel [60] discuss the case of *otaku* (i.e., an individual with consuming interests, such as anime or manga) and the Cool Japan initiative in terms of the fluidity of the meaning and interpretations of such social categories, where the original and mostly negative label *otaku* has started to decrease in perceived negativity [61]. Hence, this cluster appears to be comparing and contrasting societal perceptions of *otaku* and *hikikomori*. Similarly, the cited references also discuss several such labels and social categories such as *moe*, *soushoukukei otoko* (“herbivore”, usually used with reference to men) (e.g., [62,63]) as well as the Cool Japan initiative. This cluster indicates the juxtaposition of *hikikomori* in relation to other similar youth labels and social categories from a more sociological perspective and how this may influence research and public policy relating to *hikikomori*.

### 4.4. Cluster #7: Youth Services

The articles citing the most references in the cluster in Cluster “Youth services were authored by Chan and Lo [64] covering 20 articles and GCS of 21, Krieg and Dickie [9] covering 11 articles and GCS of 50 and Wong et al. [15] covering 4 articles and GCS of 76. The main theme of the cluster seems to centre on youth services such as programs and activities that have been designed for *hikikomori* youth, with a number of these services being cited as references in this cluster (e.g., [65,66]. Notably, the major citing article by Chan and Lo [64] reviewed and compared the available services between Hong Kong and Japan, and proposed recommendations to enhance those available in Hong Kong. Specifically, Chan and Lo [64] proposed the incorporation of more therapeutic elements with youth and their families as well as the need for diversifying the range of services available in Hong Kong. The results reported in Chan and Lo [64] suggest that there may be differing perceptions of the nature of *hikikomori* in different countries which consequently shape the types of services available. Hence, this cluster shows the utility in conducting research evaluating youth services and their efficacy in reaching out and alleviating social withdrawal in *hikikomori* across different countries which can be used as a reference for the design and implementation of a holistic range of evidence-based services.

### 4.5. Cluster #4: Censure and Eempowerment of Hikikomori

The articles citing the most references in the cluster in Cluster “Censure and empowerment of hikikomori” was authored by Chan [67] covering 20 articles and GCS of 5, Li and Wong [68] covering 15 articles and GCS of 96, and Tajan [69] covering 9 articles and GCS of 31 (Table 5). The major citing article examines the phenomenon of *hikikomori* in Hong Kong, where it is termed *hidden youth*. Using a sociological perspective, the article cites the Social Censure Theory in arguing that negative labelling is thought to be reflective of moral judgment by the dominant social class. A key argument raised in the article is the role of the negative labelling placed on *hikikomori* in Hong Kong in exacerbating a cycle of resistance where the youth eventually recognise “being hidden”-or withdrawing-as the ultimate form of resistance to the censure placed on them. The argument is in line with the cited article by Burkley and Blanton [70] of the negative outcomes with internalising negative stereotypes, such as “behavioural assimiliation” where individuals behave in ways consistent with stereotypes and low self-esteem. Accordingly, the cited articles include initiatives such as job training for NEET individuals, where participants reported more of a need for social and emotional support and a better sense of self than increasing their employability [71]. Conversely, the cited articles also include research into the internet as an avenue where *hikikomori* can regain empowerment and a sense of self [72,73]. The characteristics of the internet grants anonymity and autonomy to youths to explore their preferred self-identity and social interactions. Hence, the cluster highlights the importance of recognising societal judgments and censure placed on youths, which can inevitably shape how adults approach, design and treat *hikikomori* youth, as well as empowerment and disempowerment as key elements of the *hikikomori* narrative.

### 4.6. Cluster #29: Biological Markers of Hikikomori

The articles citing the most references in the cluster in cluster “Biological markers in hikikomori” was authored by Hayakawa et al. [74], covering 9 articles and GCS of 21. The article reported possible blood biomarkers for *hikikomori* individuals, including serum uric acid levels in men and high-density lipoprotein cholesterol in women. The biomarkers tested in the study were related to avoidant personality traits, and the cited articles focused on similar studies investigating biological markers and psychiatric disorders, including uric acid in major depressive and anxiety disorders [75] and serotonin transporter promoter polymorphism [76]. The recency of this cluster is an encouraging sign of research moving towards biological markers of *hikikomori*, which holds potential as a possible diagnostic tool and could be researched further in terms of common biological markers of *hikikomori* across cultures. A common biological marker for *hikikomori* could serve as a potential basis to consolidate research on prolonged social withdrawal across countries and serve as a part of a universally shared definition for *hikikomori*.

### 4.7. Cluster #28: Gaming as an Intervention for Hikikomori

The articles citing the most references in the cluster in cluster “Gaming as an intervention for hikikomori” were authored by Hussain [77] with a coverage of 8 articles and GCS of 5, Kato et al. [11] with a coverage of 2 articles and GCS of 60 and Tateno et al. [78] with a coverage of 2 articles and GCS of 86 (Table 6). The cluster appears to focus on the use of gaming as an intervention for *hikikomori* with the major citing article by Hussain [77] reporting a study on the mobile application game Pokemon Go. Similarly, Kato et al. [11] also mentioned the role of Pokemon Go in motivating *hikikomori* to leave their homes, a sentiment echoed by the cited article by Tateno et al. [79]. Many of the cited articles were related to previous studies that looked at concepts of “exergaming”, which combines physical activity and gaming (e.g., [80,81,82,83]). The gameplay of Pokemon Go involves physically travelling in the real world, to catch Pokemon which spawn at real world locations. The nature of this gameplay means that it holds potential in encouraging *hikikomori* to leave the confines of their homes and promoted their engagement in physical activity. Accordingly, the study by Althoff et al. [82] reported an increase in physical activity across men and women of various ages who played Pokemon Go. More research can be conducted on not only the efficacy of Pokemon Go, but also other forms of gamification involving exercise in motivating *hikikomori* to leave their homes, although it is likely that the recent COVID-19 pandemic may have stalled such research efforts.

### 4.8. Cluster #11: Sociological Perspective of Hikikomori

The articles citing the most references in the cluster in cluster “Sociological perspective of hikikomori” were authored by Caputo [85] covering 9 articles and GCS of 2, Overell [86] covering 6 articles and GCS of 1 and Kirjavainen and Jalonen [87] covering 5 articles and GCS of 2 (Table 7). The work in the cluster seems to focus on a sociological perspective of *hikikomori*, where the main citing articles discuss *hikikomori* and their relationship with mainstream society. *Hikikomori* individuals are argued by Overell [86] to be out of place in the dominantly masculine culture in Japan while the analysis of forum posts reported in the study by Kirjavainen and Jalonen [87] indicated frustration towards society and a poorly functioning job market with difficulty finding meaningful employment. Similarly, the cited articles highlight the recession and irregular labour market in Japan [88], as well as the prevailing reality of limited upward social mobility across Western societies [89]. Hence, the cluster points towards a group of work looking into the role of societal forces that promote and enable *hikikomori*, with an emphasis on limited opportunities for productive and meaningful employment.

### 4.9. Cluster #40: Identity Content Valence

In cluster “Identity content valence”, the major citing document was authored by Hihara et al. [92], covering 4 documents and GCS of 4. The document focuses on investigating the relationship between identity content valences and adaptation/maladaptation in Japanese young adults. It emerged that *hikikomori* symptoms predicted negative identity elements.

### 4.10. Cluster #17: Hikikomori Across Cultures

The articles citing the most references in the cluster in cluster “Hikikomori across cultures” were authored by Nonaka and Sakai [93] covering 6 articles and GCS of 5, De Luca [94] covering 6 articles and GCS of 5 and De Luca [95] covering 6 articles and GCS of 5. The focus of the work in this cluster seems to be research on the *hikikomori* phenomenon outside Japan, where it first gained interest, and the cultural dimensions associated with *hikikomori*. Although *hikikomori* in Japan has origins in Japanese mythology, similar reports of prolonged social withdrawal have also been made in France and in England in the 1950s and 1970s, where such French and English nosography was explored in greater detail in De Luca [94]. Moreover, with the publication of Saito’s seminal work on *hikikomori* in English in 2012 [1], reports of prolonged social withdrawal across the world began to surface as evidenced by the cited articles [12,45,96] and the use of *hikikomori* entered the lexicon of the “mainstream” international research community. Notably, *hikikomori* is included as a cultural idiom in the DSM-5 rather than a psychiatric diagnosis, which remains an ongoing debate since a sociocultural approach towards *hikikomori* is favoured in Japan [94]. Research in cluster “Hikikomori across cultures” also allude to the need for greater consensus in the global psychiatric community as to the diagnosis of *hikikomori* and prolonged social withdrawal as a psychiatric condition in and of itself.

### 4.11. Cluster #1: Etiology and Risk Factors of Hikikomori

The articles citing the most references in the cluster in cluster “Etiology and risk factors of hikikomori” were authored by Chan [67] covering 29 articles and GCS of 5, Orsolini et al. [97] covering 19 articles and GCS of 2 and Kubo et al. [98] covering 17 articles and GCS of 0 (Table 8). The main theme of this cluster appears to be risk factors and etiology of *hikikomori*. Many perspectives on the causes of *hikikomori* have been proposed including social censure [67], family environments [99,100], maternal attachment [9] but recently, there is a move towards the consideration that *hikikomori* is a society-bound syndrome rather than a cultural-bound one, likely to be the demands of modern, post-industrial societies as argued by the major citing article by Martinotti et al. [101]. Recently, researchers have proposed the similarity of *hikikomori* with a novel psychiatric syndrome, modern type depression, which made up the subject of a number of the cited articles in this cluster [102,103,104,105]. It was proposed that characteristics of modern type depression may be shared with *hikikomori* including avoidance of societal hierarchies and ranks, a preference for existence without social roles and a vague sense of omnipotence [104,105]. Moreover, many recently published citing articles also raised the relationship between *hikikomori* and modern type depression [97,101], where the major citing article authored by Kubo et al. [98] also found an association between modern type depression between *hikikomori*, where the authors proposed that both conditions may be gateways to the other.

### 4.12. Cluster #31: Hikikomori on Twitter

Cluster “Hikikomori on Twitter” had an average year of publication in 2016. The major citing document in the cluster was authored by Pereira-Sanchez et al. [111], with a coverage of 7 documents and a GCS of 17. Particularly, [111] used Twitter to explore perceptions about *hikikomori* in Western countries.

### 4.13. Cluster #8: Social Media Usage in Hikikomori

The major citing article in cluster “Social media usage in hikikomori” was authored by Bozzola et al. [112] covering 12 articles and GCS of 21, followed by Tateno et al. [78] (coverage = 10; GCS = 86) and Stavropoulos et al. [113] (coverage = 8; GCS = 38) (Table 9). Bozzola et al. [112] reports the recommendations by the Italian Pediatric Society for device use by adolescents, where they highlighted *hikikomori* as one phenomenon closely related to device use and at high risk of internet addiction [84]. Social media and gaming applications on mobile devices appear to be commonly used applications by adolescents and is likely the case for *hikikomori* youth as well. As smartphones continue to become a more integral part of everyday life and adolescents are beginning to gain access to mobile devices at an increasingly younger age. As a result, issues of smartphone addiction and internet addiction become of serious concern, especially in *hikikomori* youth, who are reportedly spending a significant time-more than 12 h of screen time [68,84]. Accordingly, the cited articles are studies investigating smartphone addiction in youths [114,115]. The study by Tateno et al. [78] found that *hikikomori* trait had a relatively strong correlation with internet addiction, where those at high-risk for *hikikomori* spent longer times using the internet. Similarly, Stavropoulos et al. [113] found that *hikikomori* symptoms were associated with internet gaming disorder—with game playing time being a moderator of this association-in a sample of massively multiplayer online game users. In the virtual world where online avatars and identities can be created, the virtual reality may be more appealing for *hikikomori* youth [84]. The proposed factors underlying gaming motivations by Yee [116], desire for interaction with others, gaming as a form of escapism from real life distress, immersing into a virtual identity and a desire for achievement which can be fulfilled in-game, may be rather consistent with *hikikomori* who may have experienced bullying or been ostracised in school or failed to meet expectations. With the versatility and functionality of the internet encompassing gaming, video streaming, social media, and online shopping to meet everyday needs, internet addiction is a very significant concern for *hikikomori* youth, which should be a key consideration in intervention design. Hence, at the same time that it is an encouraging sign that research in cluster “Gaming as an intervention for *hikikomori*” is looking into the viability of mobile applications such as Pokemon Go in engaging *hikikomori* youth, a balance should also be struck in terms of being aware of excessive usage and addiction to mobile devices in the design of such online or gaming intervention.

### 4.14. Cluster #16: Experiencing Hikikomori

The articles citing the most references in the cluster in cluster “Experiencing hikikomori” were authored by Hill [118] covering 5 documents and GCS of 2, Vainikka [119] covering 5 and GCS of 3 and by Bradley [120] covering 5 and GCS of 0 (Table 10). The major citing documents investigate the experience of people with *hikikomori*.

### 4.15. Cluster #0: Clinical Features of Hikikomori

The articles citing the most references in the cluster in cluster “Clinical features of hikikomori” were authored by Amendola et al. [122] with covering 26 articles and GCS of 1, Kato et al. [11] covering 25 articles and GCS of 60 and Li and Wong [68] covering 22 articles and GCS of 96 (see Table 11). The clinical presentation of *hikikomori* seems to be the focus of this cluster, where some of the major citing articles [122,123] were evaluating the psychometric properties of Italian versions of the *Hikikomori* Questionnaire. Moreover, other citing articles discussed the clinical features of *hikikomori* [11,101,124,125,126,127], where the cited articles included those discussing the definitions and/or diagnostic criteria of *hikikomori* [11,49]. In general, there appears to be multiple operational definitions of *hikikomori* such as those outlined in Saito [2], Teo and Gaw [45], Teo et al. [50] without a standardised clinical definition. Results from the review by Nonaka et al. [124] also suggested that there may be differences in studies conducted in Japan and the rest of the world, where the authors suggested the possible influence of the researcher’s perception of *hikikomori*. For example, the initial guidelines set out by the Japanese Ministry of Health, Labour and Welfare does not include functional impairment in its definition unlike the definitions proposed by Teo and Gaw [45] and Kato et al. [49]. Moreover, the comparison between *hikikomori* in French and Japanese adolescents conducted by Hamasaki et al. [126] found that the pathology of *hikikomori* did not differ, where both French and Japanese adolescents showed high “parental psychiatric disorders”, “overuse of Internet” and low “communication between parents”. However, “communication with the community” only contributed to *hikikomori* severity in the French sample, suggesting cultural differences in the risk factors *hikikomori*. As such, the authors propose that *hikikomori* is a common phenotype with several possible underlying psychopathological mechanisms, similar to the conclusions drawn by Kato et al. [48]. Hence, the work in the cluster suggests that more research may be needed to determine the underlying pathologies, which may require varying strategies and interventions to alleviate.

### 4.16. Cluster #44: COVID-19

Cluster “COVID-19” is the most recent cluster with an average year of publication in 2020. The major citing document in the cluster was authored by Kathirvel [130] covering 3 articles and GCS of 30. The major citing document examined the implications of the social isolation during the COVID-19 pandemic for mental health consequences.

### 4.17. Country Analysis

In the current work, the country analysis relies on the country included in the authors’ affiliation string. The main countries included in the network were mostly post-industrial societies, which is in line with the conceptualisation of *hikikomori* as a society-bound syndrome associated with the demands of a modern society. This pattern may be indicative of the prevalence of *hikikomori* being reported in these countries, thereby generating greater research interest and work conducted.

Moreover, the years of the burst seem to suggest the role of media coverage and the presence of subject matter experts in the respective countries in spurring research into *hikikomori*. In the case of the United States, the burst occurred in 2007, which may be attributable to the publication of the book *Shutting Out the Sun* on *hikikomori* by Canadian journalist Michael Zielenziger [131], effectively introducing the phenomenon to the English-speaking community. In the case of Hong Kong, the burst occurred in 2014, which is in line with the work by Chan and Lo [64], who then continued to research the phenomenon in Hong Kong youth, suggesting the role of researchers in spearheading research into *hikikomori*. In the case of Singapore, the burst occurred in the year 2017, which coincides with a *hikikomori* symposium conducted by the National University of Singapore. It is possible that the symposium promoted knowledge sharing and collaborations on research into *hikikomori*, highlighting the role of research expertise of researchers in the countries in spearheading research.

### 4.18. Limitations

There are some limitations to the scientometric analysis conducted in this study. Firstly, the DCA is a quantitative analysis of the number and pattern of citations and co-citations, and does not provide insight into the nature of the citations included in the analysis. This means that the DCA does not provide a qualitative perspective of the citation patterns and does not distinguish self-citations. Moreover, although the analysis was conducted on the vast majority of downloaded articles, there is a small percentage of data loss during the data import to CiteSpace, which may have led to the exclusion of relevant articles on *hikikomori*. Secondly, the DCA does not consider the type of article being cited such as reviews, case reports or experimental studies. Thirdly, it is important to note that the sample of documents included in the analysis reported in this study may not be exhaustive since only Scopus was used as a database for the article search and there may be articles which were not included in the current analysis.

## 5. Conclusions

Research on *hikikomori* focuses on both the prevalence and presentation of *hikikomori*, as well as articles exploring the causes of *hikikomori*. The results from the review suggest the growth in research conducted on *hikikomori* across the world may have culminated in a paradigm shift in recent years towards a multidimensional approach to understanding *hikikomori*, which is an important consideration for mental health practitioners and youth services in designing therapeutic interventions to encourage *hikikomori* individuals to leave the confines of their homes. As the *hikikomori* phenomenon continues to pose a serious public health problem in countries across the world, the results from the scientometric review point towards the need for greater consensus in terms of a standardised clinical definition of *hikikomori* and validated diagnostic tools and criteria. Moreover, with *hikikomori* being identified in more countries, findings from studies conducted in different countries should be consolidated to derive a clearer picture of the presentation of *hikikomori* and its risk factors, in order to better identify populations at risk.

## Figures and Tables

**Figure 1 ijerph-20-05657-f001:**
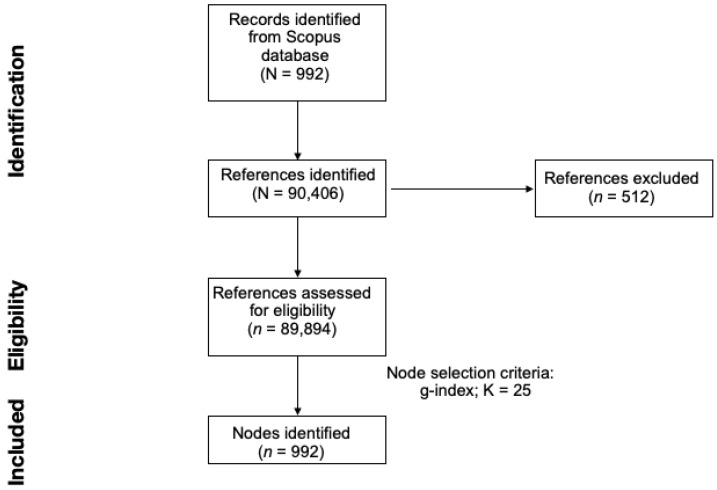
A flowchart of the PRISMA procedure employed in the study.

**Figure 2 ijerph-20-05657-f002:**
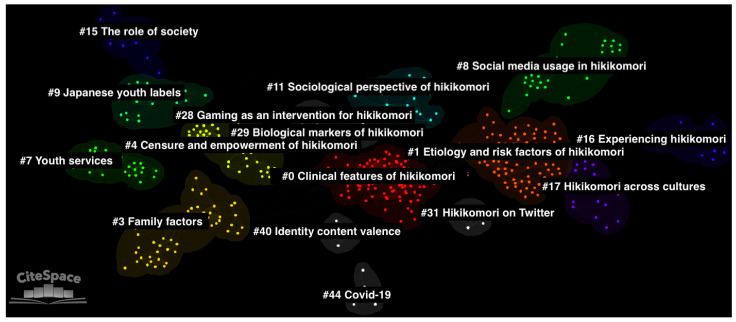
Document co-citation analysis network of all literature on the hikikomori with 16 generated clusters. In the network, single nodes represent individual documents.

**Table 1 ijerph-20-05657-t001:** Structural metrics of the 16 clusters in the network. The CiteSpace software automatically generates Log-likelihood Ratio (LLR) labels.

Cluster ID	Size	Silhouette	Mean Publication Year	LLR Label	Suggested Label
0	102	0.92	2016	Psychometric properties	Clinical features
1	82	0.88	2015	Modern-type depression	Risk factors
3	49	0.964	2007	Strategic foundation	Family factors
4	44	0.959	2011	Virtual world	Censure and empowerment of hikikomori
7	32	0.984	2009	Hong Kong	Youth services
8	28	0.978	2015	Italian pediatric society	Social media usage in hikikomori
9	26	0.95	2008	Transcending label	Japanese youth labels
11	23	0.909	2014	Transcending label	Sociological perspective of hikikomori
15	17	0.996	2001	Symbolic component	The role of society
16	16	1.000	2016	Moral experience	Experiencing hikikomori
17	16	0.992	2015	Cultural idiom	Hikikomori across cultures
28	9	1.000	2014	Preventing hikikomori	Gaming as an intervention for hikikomori
29	9	0.992	2013	Severe social withdrawal syndrome	Biological markers
31	7	0.994	1999	Extent	Hikikomori on Twitter
40	4	0.997	2015	Maladaptive functioning	Identity content valence
44	4	0.992	2020	COVID-19 pandemic mental health	COVID-19

**Table 2 ijerph-20-05657-t002:** Top 14 publications in terms of burst strength.

Reference	Citation Burstness	Publication Year	Burst Begin	Burst End	Duration	Betweenness Centrality	Sigma
Teo [44]	6.72	2010	2013	2018	5	0.05	1.35
Kato et al. [11]	5.81	2019	2020	2022	2	0.01	1.05
Furlong [10]	5.74	2008	2012	2016	4	0.00	1.02
Tateno et al. [46]	5.21	2012	2015	2019	4	0.01	1.08
Teo and Gaw [45]	5.21	2010	2013	2018	5	0.00	1.00
Kondo et al. [47]	4.66	2013	2016	2019	3	0.03	1.14
Kato et al. [48]	4.56	2018	2018	2020	2	0.03	1.15
Kato et al. [49]	4.43	2020	2020	2022	2	0.01	1.06
Teo et al. [50]	4.41	2015	2020	2022	2	0.02	1.11
Yong and Nomura [51]	4.09	2019	2020	2022	2	0.11	1.51
Krieg and Dickie [9]	3.78	2013	2016	2019	3	0.00	1.02
Chauliac et al. [13]	3.40	2017	2020	2022	2	0.01	1.03
Kato et al. [12]	3.39	2012	2018	2020	2	0.00	1.01
Saito and Angles [1]	3.14	2013	2017	2022	5	0.03	1.09

**Table 3 ijerph-20-05657-t003:** Five countries with a citation burst.

Country	Strength	Burst Begin	Burst End	Duration
France	3.33	2017	2018	1
United States	3.16	2007	2013	6
Hong Kong	2.49	2014	2018	4
Switzerland	1.93	2011	2013	2
Singapore	1.88	2018	2020	2

**Table 4 ijerph-20-05657-t004:** Major citing articles in Cluster Family Factors.

Coverage	GCS	Citing Article
15	17	Toivonen [53]
12	130	Teo and Gaw [45]
9	39	Umeda et al. [54]
5	12	Horiguchi [58]
5	99	Teo et al. [50]
4	51	Malagón-Amor et al. [14]

**Table 5 ijerph-20-05657-t005:** Major citing articles in Cluster Censure and Empowerment of hikikomori.

Coverage	GCS	Citing Article
20	5	Chan [67]
15	96	Li and Wong [68]
9	31	Tajan [69]
6	17	Li and Wong [4]

**Table 6 ijerph-20-05657-t006:** Major citing articles in Cluster Gaming as an intervention for hikikomori.

Coverage	GCS	Citing Article
8	5	Hussain [77]
2	60	Kato et al. [11]
2	86	Tateno et al. [78]
1	60	Stip et al. [84]

**Table 7 ijerph-20-05657-t007:** Major citing articles in Cluster Sociological perspective of hikikomori.

Coverage	GCS	Citing Article
9	2	Caputo [85]
6	1	Overell [86]
5	2	Kirjavainen and Jalonen [87]
3	13	Rubinstein [90]
2	5	Umemura et al. [91]

**Table 8 ijerph-20-05657-t008:** Top 10 major citing articles in Cluster Etiology and risk factors in hikikomori.

Coverage	GCS	Citing Article
29	5	Chan [67]
19	2	Orsolini et al. [97]
17	0	Kubo et al. [98]
15	2	Yung et al. [106]
14	2	Funakoshi et al. [107]
12	3	Martinotti et al. [101]
10	60	Stip et al. [84]
10	1	Ranieri [108]
9	0	Ari and Mari [109]
9	3	Masi et al. [110]

**Table 9 ijerph-20-05657-t009:** Major citing articles in Cluster Social media usage in hikikomori.

Coverage	GCS	Citing Article
12	21	Bozzola et al. [112]
10	86	Tateno et al. [78]
8	38	Stavropoulos et al. [113]
3	7	Voiskunskii and Soldatova [117]

**Table 10 ijerph-20-05657-t010:** Major citing articles in Cluster Experiencing hikikomori.

Coverage	GCS	Citing Article
5	2	Hill [118]
5	3	Vainikka [119]
5	0	Bradley [120]
3	21	Rooksby et al. [121]

**Table 11 ijerph-20-05657-t011:** Top 10 major citing articles in Cluster Clinical features of hikikomori.

Coverage	GCS	Citing Article
26	1	Amendola et al. [122]
25	60	Kato et al. [11]
22	96	Li and Wong [68]
21	3	Martinotti et al. [101]
21	31	Kato et al. [128]
20	2	Nonaka et al. [124]
18	0	Hamasaki et al. [126]
18	8	Katsuki et al. [129]
15	14	Kato et al. [103]
14	10	Hamasaki et al. [125]

## Data Availability

Not applicable.

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
