# Peer review of "Hikikomori: A Scientometric Review of 20 Years of Research"

_ijerph, 2023, doi:10.3390/ijerph20095657_

Round 1
Reviewer 1 Report
Dear Editor and Authors
I read with interest this scientometric review of studies about hikikomori. It is a timely analysis considering the increasing interest in hikikomori and social withdrawal more broadly.
After carefully reading the manuscript, I believe it has strengths, such as the use of a novel approach, and different methodological limitations. I offer some recommendations below to help the authors in improving their manuscript.
Major limitations
1) The paper does not contain a section about the aims and applications of the scientometric approach in reviewing the literature on a topic of interest. It could be added at the beginning of the Materials and Method section to inform the readers about its uses.
2) The usefulness of this study is missing. Why do we need it? What may it add to scientific knowledge?
3) Usually, the above is described at the end of the introduction section together with the study aims and hypotheses. Considering that they are lacking, I suggest that the authors clarify the explorative nature of this study.
4) P:2, Line 80: “When articles are frequently cited together, it can be assumed that they reflect shared research trends and intellectual domains”. Another explanation accounting for some of the variance in the fact that articles may be cited together is self-citation. Could articles co-citations include self-citation and then reflect, in part, intellectual conflict of interests? Are self-citations taken into account/controlled for in the analysis? If yes, how? Please clarify. Otherwise, this constitutes a major limitation of the study.
5) P. 3, Line 82: “In order to obtain a balanced network of documents, a parameter optimisation was conducted by computing and comparing several DCAs, each with a different setting for one of three node selection criteria; g-index, TOP N, TOP N%”. I recommend that the authors present all those results as supplementary materials (also for the country analysis).
6) Line 87: G-index: The description is not clear, for example "represents the largest value" (largest value of what?). Further, "average number of citations of the mostly highly cited publications" (what does it mean? Does it consider all publications of an author or only cited publications?).
7) Line 95: “g-index with scale factor k set at 25, 50 and 75, TOP N with scale factor N set at 50, 100 and TOP N% with scale factor N set at 10”. What do these values represent? Were they arbitrarily used? Please clarify. For example, a previous study used g-index=30 (10.1371/journal.pone.0223994). From Wikipedia: "The g-index is an author-level metric suggested in 2006 by Leo Egghe.[1] The index is calculated based on the distribution of citations received by a given researcher's publications, such that given a set of articles ranked in decreasing order of the number of citations that they received, the g-index is the unique largest number such that the top g articles received together at least g2 citations. Hence, a g-index of 10 indicates that the top 10 publications of an author have been cited at least 100 times (10 to the square), a g-index of 20 indicates that the top 20 publications of an author have been cited 400 times (20 to the square). It can be equivalently defined as the largest number n of highly cited articles for which the average number of citations is at least n.". Regarding this manuscript, it means that a g-index of 25 indicates that the top 25 publications of an author have been cited at least 625 times (25 to the square). But is the above realistic? How many authors published 25 or more publications on hikikomori? Could have been informative/better to use g-index values lower than 25? Please clarify. Indeed, it is especially important to the decision of the values of scale factors. If erroneous, the results are unreliable.
8. Cluster labelling procedure: Labelled according to which parameters or characteristics? Please provide information that supports cluster labels (for all clusters for example in a table). Otherwise, it seems subjective and untrustworthy. Clarify the procedure in the Methods section.
9. LLR label: Please discuss them in the results section as well as in Methods section.
10. Table 1: Cluster ID: Why some clusters are missing/not reported? How many clusters were identified? Including a figure showing them is necessary.
11. I will consider more specifically both the Results and Discussion sections during the second stage of the revision of this manuscript. Some initial advice regards the inclusion of a study limitations section and the discussion of country analysis results (as done for clusters resulting from the first analysis).
Minor limitations
12. P. 1, line 17: “largely”. What does it mean? Please clarify. The authors would agree that despite the proposal of Kato et al. (2019, 2020) for the hikikomori criteria, a consensus has not been reached by researchers on its definition. Therefore, it is necessary to be cautious in describing hikikomori. For example, the Cabinet Office of Japan considers hikikomori as those individuals that go out in the neighbourhood (e.g., to the convenience store), those that leave their own room, but not the house and those that rarely leave their own room. However, Kato et al. have specified an “arbitrary” number of days in going out vs. not doing so to identify an individual as hikikomori. Therefore, a clarification on the meaning of “largely” is needed to avoid further vagueness.
13. P. 2, line 60 and 74: “as (done) in …” The in-text references to the authors of the studies are lacking.
14. Line 72: Are invalid references "negligible"? How was it estimated? Could the references considered invalid be relevant articles on hikikomori? If not the focus of a specific analysis, this may constitute a study limitation (because of potential influences on the results).
15. Line 90: “1 year”. Which one is the intended period? Does it refer to the 1-year after the publication of an article? Clarify.
16. Line 108: please clarify what a value of 0 vs 1 means (as done for Silhouette scores).
17. Line 116: “groundbreaking and revolutionary”. Clarify in which way, otherwise, it seems that it represents the N times that a paper has been cited concomitantly to other papers on the same topic (hikikomori in this case). In my view, a statistical analysis cannot indicate if a work is "groudbreaking or revolutionary" (only the interpretation of the results – and then the researcher(s) - does).
18. Line 120: “given time frame”: Which time frame? Please clarify.
19. Line 121: “novelty”: I understand that sigma represents the influence of an article, but I do not understand in which way it indicates its "novelty". Please clarify. Otherwise, remove it.
20. Line 134: “major clusters”: What does it mean "major cluster"? and what does it qualify a cluster as "major"? Please clarify in Methods section.
21. Same line: figure is lacking.
22. Line 146: “after duplicates of the same documents were omitted”. What does it mean? Please clarify in the text.
23. Once the clusters’ labels are decided, please refer to each cluster with its name in text rather than using number(s).
24. In results section: please discuss the (very low) values of betweenness centrality. This may be a sign of an error in the statistical procedure.
25. Line 154: “when ʏ = 0.60”. What does it mean? Were different analyses according to different values performed? Clarify in the text.
Author Response
Dear Editor and Authors
I read with interest this scientometric review of studies about hikikomori. It is a timely analysis considering the increasing interest in hikikomori and social withdrawal more broadly. After carefully reading the manuscript, I believe it has strengths, such as the use of a novel approach, and different methodological limitations. I offer some recommendations below to help the authors in improving their manuscript.
>> We thank the Reviewer for the comments. In order to further improve the manuscript, we have updated the analysis/results by using the newest version of the software.
Major limitations
[R1.1] The paper does not contain a section about the aims and applications of the scientometric approach in reviewing the literature on a topic of interest. It could be added at the beginning of the Materials and Method section to inform the readers about its uses.
>> We thank the Reviewer for the comment. We have added the aims of the current study at the end of the Introduction section as a subsection for clarity (Lines 47-63).
[R1.2] The usefulness of this study is missing. Why do we need it? What may it add to scientific knowledge?
>> We clarified the usefulness of the current study at the end of the Introduction section (Lines 47-63).
[R1.3] Usually, the above is described at the end of the introduction section together with the study aims and hypotheses. Considering that they are lacking, I suggest that the authors clarify the explorative nature of this study.
>> We have outlined more clearly the aim and the nature of the scientometric review in the subsection on the aims of the study as mentioned above (Lines 47-63).
[R1.4] P:2, Line 80: “When articles are frequently cited together, it can be assumed that they reflect shared research trends and intellectual domains”. Another explanation accounting for some of the variance in the fact that articles may be cited together is self-citation. Could articles co-citations include self-citation and then reflect, in part, intellectual conflict of interests? Are self-citations taken into account/controlled for in the analysis? If yes, how? Please clarify. Otherwise, this constitutes a major limitation of the study.
>> Self co-citations could be represented in the network. To control these types of bias, all documents were initially screened by the software using the node selection criteria and scale factor. In this way, only the most frequently cited documents in the literature were included in the network and were used for the co-citation analysis. Although we can apply these strategies to prevent these types of bias, we cannot distinguish between an intellectual conflict of interest or a common thematic interest/approach by the research groups in terms of the nature of citations in work investigating hikikomori. We have mentioned this in the limitations (Lines 478-480).
[R1.5] P. 3, Line 82: “In order to obtain a balanced network of documents, a parameter optimisation was conducted by computing and comparing several DCAs, each with a different setting for one of three node selection criteria; g-index, TOP N, TOP N%”. I recommend that the authors present all those results as supplementary materials (also for the country analysis).
>> We have provided the results for all the node selection criteria and scale factors used in the Results section of the manuscript (Lines 146-154; Lines 181-183).
[R1.6] Line 87: G-index: The description is not clear, for example "represents the largest value" (largest value of what?). Further, "average number of citations of the mostly highly cited publications" (what does it mean? Does it consider all publications of an author or only cited publications?).
>> We clarified the meaning of the g-index in the text (Lines 83-85).
[R1.7] Line 95: “g-index with scale factor k set at 25, 50 and 75, TOP N with scale factor N set at 50, 100 and TOP N% with scale factor N set at 10”. What do these values represent? Were they arbitrarily used? Please clarify. For example, a previous study used g-index=30 (10.1371/journal.pone.0223994). From Wikipedia: "The g-index is an author-level metric suggested in 2006 by Leo Egghe.[1] The index is calculated based on the distribution of citations received by a given researcher's publications, such that given a set of articles ranked in decreasing order of the number of citations that they received, the g-index is the unique largest number such that the top g articles received together at least g2 citations. Hence, a g-index of 10 indicates that the top 10 publications of an author have been cited at least 100 times (10 to the square), a g-index of 20 indicates that the top 20 publications of an author have been cited 400 times (20 to the square). It can be equivalently defined as the largest number n of highly cited articles for which the average number of citations is at least n.". Regarding this manuscript, it means that a g-index of 25 indicates that the top 25 publications of an author have been cited at least 625 times (25 to the square). But is the above realistic? How many authors published 25 or more publications on hikikomori? Could have been informative/better to use g-index values lower than 25? Please clarify. Indeed, it is especially important to the decision of the values of scale factors. If erroneous, the results are unreliable.
>> As outlined in the general guidelines of the software, scale factors for node selection criteria should be optimised to obtain a balanced network based on the structural properties of the network and the number of identified major clusters. The above mentioned method to include documents would not be realistic and, in CiteSpace, it is represented by the case in which the scale factor k is set to k = 1. For enabling the users to control the overall size of the final network, CiteSpace implements a modified version of the g-index with a scale factor k. Higher values of k correspond to a higher number of included documents. Therefore, the scale factor k sets the threshold of the g-index criterion. We clarified this in the manuscript in Lines 102-106.
[R1.8] Cluster labelling procedure: Labelled according to which parameters or characteristics? Please provide information that supports cluster labels (for all clusters for example in a table). Otherwise, it seems subjective and untrustworthy. Clarify the procedure in the Methods section.
>> Labels were initially automatically generated by means of the log-likelihood ratio method, which is implemented in CiteSpace and it provides the most accurate results. The authors conducted a qualitative inspection of clusters and, where appropriate, amended the LLR label. A table with the main citing documents was added for the clusters with more than three citing documents. For the other clusters, all citing documents are discussed and their metrics are provided in the main text.
[R1.9] LLR label: Please discuss them in the results section as well as in Methods section.
>> We have discussed the LLR labels in the Methods and Results sections.
[R1.10] Table 1: Cluster ID: Why some clusters are missing/not reported? How many clusters were identified? Including a figure showing them is necessary.
>> CiteSpace software automatically reports only the major clusters in the network based on their metrics. The clusters that were not reported in the table were the ones that CiteSpace software did not include as major clusters in the literature. We have specified this aspect in Lines 157-158. Moreover, we have now included a figure of the network as Figure 2.
[R1.11] I will consider more specifically both the Results and Discussion sections during the second stage of the revision of this manuscript. Some initial advice regards the inclusion of a study limitations section and the discussion of country analysis results (as done for clusters resulting from the first analysis).
>> Thank you for your comments. We have included a limitations section (Lines 475-484) and a subsection discussing the country analysis (Lines 456-474).
Minor limitations
[R1.12] P. 1, line 17: “largely”. What does it mean? Please clarify. The authors would agree that despite the proposal of Kato et al. (2019, 2020) for the hikikomori criteria, a consensus has not been reached by researchers on its definition. Therefore, it is necessary to be cautious in describing hikikomori. For example, the Cabinet Office of Japan considers hikikomori as those individuals that go out in the neighbourhood (e.g., to the convenience store), those that leave their own room, but not the house and those that rarely leave their own room. However, Kato et al. have specified an “arbitrary” number of days in going out vs. not doing so to identify an individual as hikikomori. Therefore, a clarification on the meaning of “largely” is needed to avoid further vagueness.
>> Thank you for your comments. We have removed the term “largely” to eliminate the ambiguity.
[R1.13] P. 2, line 60 and 74: “as (done) in …” The in-text references to the authors of the studies are lacking.
>> We added the in-text references in Line 65 and 80.
[R1.14] Line 72: Are invalid references "negligible"? How was it estimated? Could the references considered invalid be relevant articles on hikikomori? If not the focus of a specific analysis, this may constitute a study limitation (because of potential influences on the results).
>> The invalid references are identified by CiteSpace software when importing the data collected from Scopus. During scientometric studies, a small percentage of data loss is common and it means that there were some errors in the citation format of some documents. Some references could be relevant articles on hikikomori. Even when considering that there was a percentage of data loss, the analyses were still conducted on the patterns of co-citations on a data sample of 11,373 out of a total of 11,681 cited references. We clarified this aspect in the study limitations (Lines 480-483).
[R1.15] Line 90: “1 year”. Which one is the intended period? Does it refer to the 1-year after the publication of an article? Clarify.
>> It means that the node selection criteria and scale factors were applied on a year-by-year basis to retrieve the maximum amount of information from the data sample. We have clarified this in the text in Line 96-99.
[R1.16] Line 108: please clarify what a value of 0 vs 1 means (as done for Silhouette scores).
>> We clarified the meanings of the values (Line 128).
[R1.17] Line 116: “groundbreaking and revolutionary”. Clarify in which way, otherwise, it seems that it represents the N times that a paper has been cited concomitantly to other papers on the same topic (hikikomori in this case). In my view, a statistical analysis cannot indicate if a work is "groudbreaking or revolutionary" (only the interpretation of the results – and then the researcher(s) - does).
>> We have replaced “groundbreaking and revolutionary” with “high impact work” in Line 128 of the manuscript.
[R1.18] Line 120: “given time frame”: Which time frame? Please clarify.
>> The time frame represents the duration of the citation burstness identified by the software through Kleinberg's algorithm. Duration of bursts represent a result and are described in Table 2 and in the section 3.3 of the manuscript.
[R1.19] Line 121: “novelty”: I understand that sigma represents the influence of an article, but I do not understand in which way it indicates its "novelty". Please clarify. Otherwise, remove it.
>> Sigma is a standard metric used in the bibliometric literature to assess the novelty of a document. To do so, it considers together citation burstness, which is a structural metric, and betweennes centrality, which is a temporal metric. The combination of structural and temporal metrics allows sigma to represent the novelty of a node. We clarified this in the manuscript in Lines 133-136.
[R1.20] Line 134: “major clusters”: What does it mean "major cluster"? and what does it qualify a cluster as "major"? Please clarify in Methods section.
>> Major clusters are automatically identified by CiteSpace based on the metrics of the clusters. We clarified this in Lines 157-158
[R1.21] Same line: figure is lacking.
>> We have now added the figure of the network as Figure 2.
[R1.22] Line 146: “after duplicates of the same documents were omitted”. What does it mean? Please clarify in the text.
>> We have clarified in the text in Lines 171-172.
[R1.23] Once the clusters’ labels are decided, please refer to each cluster with its name in text rather than using number(s).
>> In the Discussion of the revised manuscript, we refer to clusters by their label rather than by ID.
[R1.24] In results section: please discuss the (very low) values of betweenness centrality. This may be a sign of an error in the statistical procedure.
>> We conducted the analysis again by using the newest version of the software (6.1.R6). The values of betweenness centrality are low and suggest that no document is highly influential in the overall network of documents. We have specified this aspect in Lines 176-178 of the revised manuscript.
[R1.25] Line 154: “when ʏ = 0.60”. What does it mean? Were different analyses according to different values performed? Clarify in the text.
>> The gamma parameter sets the threshold for the identification of citation bursts. The default value of gamma is 1. The lower the parameter, the easier it is for a document to obtain a citation burst. We lowered the threshold in order to obtain a reasonable sample of documents with a citation burst. We clarified this in Lines 185-187.
Reviewer 2 Report
Thank you for the opportunity to review the Systematic Review “Hikikomori: A Scientometric Review of 20 years of Research”. The paper addresses an interesting and well researched theme in the recent period about the implications, challenges and difficulties of defining the analysed concept of social withdrawal in many countries. The paper is submitted under the topic “The Causes, Counseling and Prevention Strategies for Maladaptive and Deviant Behaviors in Schools”.
This study represents a new approach in the field, discussing the subject that needs to be comprehensively analyzed because “the scientometric review indicates many perspectives on the etiology of hikikomori including cultural, attachment, family systems and sociological approaches”, as the authors underline.
Also, the study is written in an adequate manner and the results are presented clearly and coherently, using visuals and text. The tables presented in the paper were relevant to explore the results of the systematic review and the ways these were adapted to the explanations in the text.
As the authors say, one conclusion assumes that this systematic review points “towards the need for greater consensus in terms of a standardised clinical definition of hikikomori and validated diagnostic tools and criteria”.
Moreover, there are some observations that should be addressed in this revision.
- The main objective of this study “aimed to identify key publications and trends in the research on hikikomori, the focus of these publications and gaps in the literature” (Line 51-52) should be presented point-by-point in the Conclusions section.
- After describing the clusters and their specificities, one figure should be added to visually link all the data presented in the Discussion section, about the cited articles, the coverage, etc.
- Correct all the typos, e.g. Line 155: “An total of 5 countries”.
Author Response
Thank you for the opportunity to review the Systematic Review “Hikikomori: A Scientometric Review of 20 years of Research”. The paper addresses an interesting and well researched theme in the recent period about the implications, challenges and difficulties of defining the analysed concept of social withdrawal in many countries. The paper is submitted under the topic “The Causes, Counseling and Prevention Strategies for Maladaptive and Deviant Behaviors in Schools”. This study represents a new approach in the field, discussing the subject that needs to be comprehensively analyzed because “the scientometric review indicates many perspectives on the etiology of hikikomori including cultural, attachment, family systems and sociological approaches”, as the authors underline. Also, the study is written in an adequate manner and the results are presented clearly and coherently, using visuals and text. The tables presented in the paper were relevant to explore the results of the systematic review and the ways these were adapted to the explanations in the text. As the authors say, one conclusion assumes that this systematic review points “towards the need for greater consensus in terms of a standardised clinical definition of hikikomori and validated diagnostic tools and criteria”.
>> We thank the Reviewer for the comments. In order to further improve the manuscript, we have updated the analysis/results by using the newest version of the software.
Moreover, there are some observations that should be addressed in this revision.
[R2.1] The main objective of this study “aimed to identify key publications and trends in the research on hikikomori, the focus of these publications and gaps in the literature” (Line 51-52) should be presented point-by-point in the Conclusions section.
>> We thank the Reviewer for the suggestion. We have now presented the Conclusions section accordingly.
[R2.2] After describing the clusters and their specificities, one figure should be added to visually link all the data presented in the Discussion section, about the cited articles, the coverage, etc.
>> We have now added a figure in which we present the network as Figure 2.
[R2.3] Correct all the typos, e.g. Line 155: “An total of 5 countries”.
>> We have now corrected the identified typos.
Round 2
Reviewer 1 Report
Dear Editor and Authors
I am glad to have the opportunity to review the second version of the manuscript.
The authors did substantial and careful work of revision taking into account the reviewers’ comments.
I believe that the quality and clarity of some sections (i.e., introduction and study aim, materials and methods) have been improved. Further, the inclusion of a limitations paragraph allows readers to carefully consider the potential effect of self-citations. This is especially important when a new topic is studied in the literature. Finally, well done with the discussion and conclusions sections.
Overall, I believe that this manuscript could make a significant contribution to the literature. However, I have some concerns. Please find my comments below.
I recommend the authors improve the transparency about the choices they made. When revising a manuscript, all changes to the text (i.e., what needs to be modified/removed from the text of the new version, and what needs to be added to it) should be clearly reported in the new version (e.g., using track changes in Word). In other words, all changes made to the first submitted version of the manuscript should be clear. In this case, however, the authors removed some parts of the text and highlighted in yellow the new ones. This makes the work of the reviewers very hard because they need to compare row by row the two versions (the first one and the revised one) of the manuscript. Therefore, such a method decreases the clarity of the paper. Please follow this comment for the new version/revision.
1. Further, the authors in replying to my first comment wrote “In order to further improve the manuscript, we have updated the analysis/results by using the newest version of the software.” without a reasonable explanation for such a change. Considering that it influences an essential part of the manuscript (statistical analysis, results, and discussions), it is unusual to modify statistical methods after a paper is submitted without this being asked/recommended during the review process. Importantly, some results of the first and second versions of the manuscript/analysis differ. Therefore, for clarity and to advance the consideration of this manuscript for potential publication, the authors need to
1.1) provide an explanation(s) for such a decision,
1.2) report all results of the first analysis as supplementary material,
1.3) highlight changes in the results between the two analyses in a paragraph at the end of the results section, specifying whether different statistical procedures were used for the two analyses,
1.4) and discuss possible reasons for differences in the results.
2. In the reply to comment n24 it is stated that “values of betweenness centrality are low and suggest that no document is highly influential in the overall network of documents.”. I believe that this result merits further attention in the discussion section and possible explanations necessitate being discussed.
3. Results - Country analysis paragraph: What does “country” information refer to (e.g., recruitment place of the sample, place where the study was conducted, or the publisher location) ? Please specify in the manuscript. Findings should be discussed accordingly.
4. Directly related to comment n3: I am aware of only one study published in Switzerland about hikikomori. To improve clarity, I believe that the authors need to report the list of studies belonging to each cluster (thematic and country clusters) as supplementary material. This would substantially improve the reliability of the results.
5. Discussion - lines 197-201: “Cluster #31: Hikikomori on Twitter Cluster “Hikikomori on Twitter" was the earliest cluster in the network, with an average year of publication in 1999. The major citing document in the cluster was authored by Pereira-Sanchez et al. [53], with a coverage of 7 documents and a GCS of 17. Particularly, [53] used Twitter to explore perceptions about hikikomori in Western countries.”. The average year of the publications belonging to this cluster is 1999. However, Twitter has been launched in 2006 and the article by Pereira-Sanchez was published in 2019. According to my opinion, there could be something wrong here. Am I missing something? Further, does “major citing document” mean that the article was the one citing more documents or was the one most cited by other works? Clarifications are needed on all the above. Following the procedure recommended in comment n4 will illuminate the results.
6. Despite Scopus being a comprehensive database, it is not the only one used for reviews. Generally, other important sources are searched (e.g., PubMed, PsychArticles and PsychInfo). This should be added as a study limitation.
Kind regards
Author Response
Dear Editor and Authors
I am glad to have the opportunity to review the second version of the manuscript. The authors did substantial and careful work of revision taking into account the reviewers’ comments. I believe that the quality and clarity of some sections (i.e., introduction and study aim, materials and methods) have been improved. Further, the inclusion of a limitations paragraph allows readers to carefully consider the potential effect of self-citations. This is especially important when a new topic is studied in the literature. Finally, well done with the discussion and conclusions sections.
Overall, I believe that this manuscript could make a significant contribution to the literature. However, I have some concerns. Please find my comments below.
I recommend the authors improve the transparency about the choices they made. When revising a manuscript, all changes to the text (i.e., what needs to be modified/removed from the text of the new version, and what needs to be added to it) should be clearly reported in the new version (e.g., using track changes in Word). In other words, all changes made to the first submitted version of the manuscript should be clear. In this case, however, the authors removed some parts of the text and highlighted in yellow the new ones. This makes the work of the reviewers very hard because they need to compare row by row the two versions (the first one and the revised one) of the manuscript. Therefore, such a method decreases the clarity of the paper. Please follow this comment for the new version/revision.
>> We have now used the highlighter to signal added sentences and the strikethrough function to signal removed parts. In the previous version of the manuscript, all modifications were done by following the Reviewers’ comments and, in doing so, we followed the journal’s guidelines.
- Further, the authors in replying to my first comment wrote “In order to further improve the manuscript, we have updated the analysis/results by using the newest version of the software.” without a reasonable explanation for such a change. Considering that it influences an essential part of the manuscript (statistical analysis, results, and discussions), it is unusual to modify statistical methods after a paper is submitted without this being asked/recommended during the review process. Importantly, some results of the first and second versions of the manuscript/analysis differ. Therefore, for clarity and to advance the consideration of this manuscript for potential publication, the authors need to
1.1) provide an explanation(s) for such a decision,
>> For the revised version of the manuscript, we used the new version of the software while maintaining the same analytical approach. The version of the software used for conducting the original analysis expired on 31 December 2022 and it was not possible to access it anymore after that date. To be able to ameliorate the manuscript according to the Reviewers’ comments (e.g., add the figure with the network, provide results for all optimization trials, etc.), we had to purchase the newest version of the software and re-run the same analysis with it.
1.2) report all results of the first analysis as supplementary material,
>> We provided the results from the older version of CiteSpace in the supplementary materials of the manuscript.
1.3) highlight changes in the results between the two analyses in a paragraph at the end of the results section, specifying whether different statistical procedures were used for the two analyses,
>> We have provided a new paragraph at the end of the results section to discuss differences between the results obtained from the two versions of the software.
1.4) and discuss possible reasons for differences in the results.
>> We discuss possible explanations for the differences in results by referring to the algorithm used by the adopted software.
- In the reply to comment n24 it is stated that “values of betweenness centrality are low and suggest that no document is highly influential in the overall network of documents.”. I believe that this result merits further attention in the discussion section and possible explanations necessitate being discussed.
>> We provided an explanation for the low values for betweenness centrality in network analysis.
- Results - Country analysis paragraph: What does “country” information refer to (e.g., recruitment place of the sample, place where the study was conducted, or the publisher location) ? Please specify in the manuscript. Findings should be discussed accordingly.
>> Country refers to the country of authors’ affiliations. We have specified it in the manuscript in Lines 118 and 119.
- Directly related to comment n3: I am aware of only one study published in Switzerland about hikikomori. To improve clarity, I believe that the authors need to report the list of studies belonging to each cluster (thematic and country clusters) as supplementary material. This would substantially improve the reliability of the results.
>> We have prepared a document with the cited references in each thematic cluster to be included in the Supplementary Materials. As for the country analysis, there are no thematic clusters of citing documents and the cited reference articles.
- Discussion - lines 197-201: “Cluster #31: Hikikomori on Twitter Cluster “Hikikomori on Twitter" was the earliest cluster in the network, with an average year of publication in 1999. The major citing document in the cluster was authored by Pereira-Sanchez et al. [53], with a coverage of 7 documents and a GCS of 17. Particularly, [53] used Twitter to explore perceptions about hikikomori in Western countries.”. The average year of the publications belonging to this cluster is 1999. However, Twitter has been launched in 2006 and the article by Pereira-Sanchez was published in 2019. According to my opinion, there could be something wrong here. Am I missing something? Further, does “major citing document” mean that the article was the one citing more documents or was the one most cited by other works? Clarifications are needed on all the above. Following the procedure recommended in comment n4 will illuminate the results.
>> There was an error in the information of one of the imported cited references and the correct average year of publication is 2016. We have updated this information in the text and we have reordered the discussion of the clusters to reflect this update. The term “major citing article” refers to an article in the cluster that cites the greatest number of reference articles that are a part of that cluster. The term “cited article” refers to a reference article that is part of that cluster, where the citing articles for each cluster are reported in tables in the discussion.
- Despite Scopus being a comprehensive database, it is not the only one used for reviews. Generally, other important sources are searched (e.g., PubMed, PsychArticles and PsychInfo). This should be added as a study limitation.
>> We have included the discussion of this point in the limitations.